# Durable Mechanical Circulatory Support in Adult Congenital Heart Disease: Reviewing Clinical Considerations and Experience

**DOI:** 10.3390/jcm11113200

**Published:** 2022-06-03

**Authors:** Joshua Saef, Robert Montgomery, Ari Cedars, Wai H. Wilson Tang, Joseph W. Rossano, Katsuhide Maeda, Yuli Y. Kim, Sumeet S. Vaikunth

**Affiliations:** 1Division of Cardiovascular Medicine, Department of Medicine, Perelman School of Medicine, University of Pennsylvania, Philadelphia, PA 19104, USA; joshua.saef@pennmedicine.upenn.edu (J.S.); yuli.kim@pennmedicine.upenn.edu (Y.Y.K.); 2Division of Cardiology, Children’s Hospital of Philadelphia, 3401 Civic Center Boulevard, Philadelphia, PA 19104, USA; rossanoj@chop.edu; 3Robert and Suzanne Tomsich Department of Cardiovascular Medicine, Sydell and Arnold Miller Family Heart, Vascular and Thoracic Institute, Cleveland Clinic Foundation, Cleveland, OH 44195, USA; montgor4@ccf.org (R.M.); tangw@ccf.org (W.H.W.T.); 4Division of Cardiology, Johns Hopkins Medicine, Baltimore, MD 21287, USA; acedars1@jhmi.edu; 5Division of Cardiothoracic Surgery, Children’s Hospital of Philadelphia, 3401 Civic Center Boulevard, Philadelphia, PA 19104, USA; maedak@chop.edu

**Keywords:** congenital heart disease, mechanical circulatory support

## Abstract

Adults with congenital heart disease (ACHD) patients are one of the fastest growing populations in cardiology, and heart failure (HF) is the most common cause of morbidity and mortality amongst them. The need for advanced HF therapies in ACHD patients stands to grow substantially. The anatomic considerations for placing durable mechanical circulatory support (MCS) devices in ACHD patients often require specialized approaches. Despite this, increasing evidence suggests that durable MCS can be implanted safely with favorable outcomes in ACHD patients. Expansion of MCS use in ACHD patients is imperative to improve their clinical outcomes. Knowledge of ACHD-specific anatomic and physiologic considerations is crucial to HF programs’ success as they work to provide care to this growing population.

## 1. Introduction

Successes in managing complex congenital heart disease (CHD) have driven a surge in prevalence [1]. Among CHD patients, heart failure (HF) causes substantial morbidity [2,3] and is the most common cause of mortality in adults with congenital heart disease (ACHD) [4]. Management of ACHD-related HF is complicated by the fact that anatomic complexity in the ACHD population has increased over time [5]. These factors confront HF and ACHD specialists with novel challenges as they work to allocate advanced HF therapies, including durable mechanical circulatory support (MCS) to ACHD patients.

Current United Network for Organ Sharing (UNOS) heart transplant allocation criteria prioritize MCS to qualify for higher priority listing status [6]. The anatomic considerations for placing temporary and durable MCS in ACHD patients are complex and require focused expertise. Consequently, a lower proportion of ACHD patients are listed as high-priority transplant waitlist statuses compared to patients with other cardiomyopathies. This discrepancy has resulted in ACHD patients experiencing longer waitlist times and higher waitlist mortality rates and highlights the need to further define the potential for durable MCS in ACHD patients [7,8].

### General Considerations

Assessing the severity of congenital heart disease requires several considerations, including patients’ native cardiac anatomy, surgical history, current physiology, and comorbidities. Consequently, many centers have developed dedicated multidisciplinary programs. This approach has shown clinical benefit to ACHD patients and is necessary when considering patients for MCS [9].

The 2018 American College of Cardiology/American Heart Association guidelines for the management of ACHD advocate for an anatomic and physiologic classification scheme meant to categorize disease severity using factors known to have prognostic value [10]. This new classification system represents a major advance in ACHD because anatomic complexity and clinical status frequently correlate poorly, and the natural history of many lesions is less clear with newer advances in care. The physiologic considerations within this classification include arrhythmia, concomitant valvular heart disease, exercise capacity, pulmonary hypertension, hypoxemia or cyanosis, vascular stenosis and NYHA functional classification.

While this is an unsurprising list, the manner in which ACHD patients present with the aforementioned issues is often quite different from those with acquired heart disease. Hosseinpour et al. classify patients with HF and congenital heart disease into three categories: those with uncorrected defects (e.g., a patient with an ostium secundum atrial septal defect that goes undiagnosed into adulthood), those with previous reparative surgery but with deteriorating ventricular function (e.g., a patient with dextro-transposition of the great arteries (d-TGA) who underwent an atrial switch procedure in his infancy who presents in adulthood with systemic right ventricular failure), and those with previous palliative surgery which is now failing (e.g., a patient with complex congenital heart disease palliated to a single ventricle circulation) [11]. When considering the adult population, these all speak to processes that drive clinical deterioration insidiously from birth. Thus, many ACHD patients adapt to their exertional limitations, such that they may not report symptoms until complications of HF are extreme and perhaps even irreversible [12,13].

The initial approach to care for ACHD patients with HF shares several principles with that of care for adult HF patients without CHD. Patients should be assessed regularly for clinical signs of congestion and functional decline, and providers should have low thresholds to refer patients to an ACHD provider for assessment and diagnostic studies tailored to the patient’s specific congenital heart disease.

While HF, both with and without CHD, is a state of neurohormonal perturbation, there is limited data to support elements of the guideline-directed medical therapy in CHD patients. Anatomic limitations often preclude contemporary measures for cardiac resynchronization therapy. Nevertheless, care is often focused on providing maximally tolerated medical therapy based on extrapolation from data in non-CHD patients and on treating CHD-specific contributors to poor cardiac output, such as arrhythmia or valvular dysfunction.

Exact clinical thresholds to trigger consideration for advanced HF therapies are still being defined, though current guidelines recommend consultation with ACHD and HF specialists when ACHD patients experience clinical HF or severe ventricular dysfunction [10,14]. Purported criteria have included multiple HF hospitalizations within one year, severe objective deterioration in exercise tolerance, persistent NYHA functional class III-IV symptoms, intolerance to further guideline-directed pharmacotherapy, refractory arrhythmias, and cardiac cachexia [15]. ACHD patients may experience these signs of deterioration in a less predictable fashion than those without congenital heart disease, and care should be taken for prompt recognition and evaluation should these concerns arise.

## 2. Experience with Durable Mechanical Circulatory Support in ACHD Patients

While experience with implanting durable MCS devices into adult HF patients without congenital heart disease has expanded substantially, experience is more limited in ACHD patients. A recent analysis of the Interagency Registry for Mechanically Assisted Circulatory Support (INTERMACS) database showed that of 16,182 patients who underwent durable MCS between 23 June 2006 and 31 December 2015, 126 (<0.1%) were ACHD patients [16]. Amongst them, 36% had a systemic right ventricle (morphologic right ventricle supporting the systemic circulation), and 13% had single ventricle physiology.

Similarly noteworthy is that of the 158 centers contributing to the INTERMACS registry, there were only 59 implanted durable MCSs in ACHD patients. The substantial majority (>70%) of those centers only implanted devices in one or two ACHD patients. The volume of ACHD patients on MCS tended to be higher (five or six patients) in centers with higher volumes of durable MCS in non-ACHD patients, indicating that experience with advanced HF therapies in the broader adult population is an important contributor to comfort in providing them to ACHD patients.

ACHD patients supported with MCS were younger, more often had depressed right ventricular function, and less often had depressed left ventricular function or mitral regurgitation than other adult HF patients. They were also, unfortunately, more likely to be allo-sensitized and have unfavorable chest anatomy for heart transplantation. Nonetheless, ACHD patients were more likely to have undergone device implantation with a plan for a bridge to transplant rather than as destination therapy compared to patients without congenital heart disease.

Unsurprisingly, device preferences have evolved as technology has advanced. Before 2010, HeartMate and HeartMate II were the most frequently implanted devices. The HeartWare HVAD was the most commonly reported ventricular assist device (VAD) implanted in ACHD patients between 2010 and 2017 in a recent systematic review (though more recent experience with the HeartMate 3 device was not included) [17]. Data on the use of the HeartMate 3 in ACHD is of great interest, as the device’s profile is more forgiving with complex anatomy and chest closure. Although numbers remain small, there has been a trend toward a higher proportion of biventricular assist devices and total artificial heart (TAH) being used in ACHD patients compared to non-ACHD patients. A higher proportion of ACHD patients with a systemic left ventricle underwent TAH placement (12.7%) than with a systemic right ventricle (4.4%) or a single ventricular circulation (5.9%), likely representing a higher prevalence of subpulmonic right ventricular failure in these patients [16].

There are several important considerations in ACHD patients which are essential to perioperative management, the foremost being a thorough assessment of cardiac anatomy. Moreover, many patients with congenital heart disease have abnormal pulmonary hemodynamics due to compensatory remodeling or, by design, in the context of single ventricle circulations. A thorough right heart catheterization with an assessment of pulmonary vascular anatomy and resistance offers crucial anticipatory guidance for postoperative hemodynamics. ACHD patients, particularly patients with single ventricle circulation, are also at increased risk of liver disease, as they often suffer chronically elevated systemic venous pressures from cavo-pulmonary connections [12]. Liver cirrhosis is associated with increased mortality in patients undergoing cardiac surgery [18].

Data on the postoperative management experience with durable MCS in ACHD patients are limited. A case series of six ACHD patients (among the highest volumes described in the INTERMACS registry) outlined an institutional strategy for guidance in the postoperative period. All patients received initial inotropic support for subpulmonic circulatory failure. Hypoxemia was tolerated in patients who had known right-to-left shunting or veno-venous collaterals. Patients were given pulmonary vasodilator therapy with inhaled and intravenous agents until extubation, followed by oral therapy. Fluid congestion was common in the postoperative period and postoperative nephrology consultation was requested for all patients. All patients required diuresis and two-thirds required temporary ultrafiltration or hemodialysis. Five patients survived to discharge, all of whom achieved NYHA functional class III status before discharge, and the average length of stay was over 80 days [19].

The most robust data on long term outcomes with durable MCS comes from a comparison of ACHD patients within the INTERMACS registry to propensity-matched controls. Cedars et al. evaluated early and late adverse event rates after MCS implantation, as well as changes in functional status and quality of life. Compared to matched controls, ACHD patients were more likely to experience early renal dysfunction, early and late hepatic dysfunction, early and late respiratory failure, late cardiac arrhythmia, and infection. Many of these adverse events were found to be severe, with 60% of ACHD patients with renal dysfunction requiring dialysis and 47% with arrhythmia experiencing a sustained ventricular arrhythmia requiring cardioversion. ACHD patients also experienced a higher overall mortality after MCS implant, largely driven by differences seen in the first five months post-procedurally. Multivariate analysis showed that factors associated with mortality hazards were an age greater than 50 years and a biventricular assist device or TAH implantation. Notably, the presence of a systemic right, left, or single ventricle did not affect mortality after durable MCS implantation. Nonetheless, surviving ACHD patients experienced similar improvements in all functional status and quality of life parameters to their matched non-ACHD controls [20].

Though they are associated with worse outcomes, biventricular assist devices and TAH are an important consideration in ACHD as many types of congenital heart disease leave patients with subpulmonic ventricular dysfunction and pulmonary hypertension. Thus, the proportion of ACHD patients who were implanted with these devices was three times higher than in non-ACHD patients [16]. Whether the increased mortality hazard is a consequence of the complications related to the pulmonic circulatory system is unclear. Next-generation durable biventricular or hybrid devices create unique opportunities for ACHD patients.

## 3. ACHD-Specific Anatomic Considerations

While there is great interest in offering durable MCS therapies to ACHD patients, there are practical limitations that prevent wide adoption. Most devices have been designed to support a morphologic left ventricle in patients with levocardia and biventricular physiology, and there are many congenital heart lesions with a systemic left ventricle where patients are at risk of heart failure. Examples include tetralogy of Fallot, atrio-ventricular septal defects, and Ebstein anomaly, among others. Nevertheless, the cardiac anatomic heterogeneity is immense within the ACHD population, even before considering the various surgical palliations and thoracic vascular anomalies. These factors, compounded by the need for adhesiolysis in many patients with multiple prior sternotomies and comorbidities, create technical challenges when planning MCS implantation even in the above lesions, which are most analogous to typical biventricular circulations.

Two anatomic categories that are specific to congenital heart disease and pose particular challenges when considering MCS therapies are lesions with systemic right ventricles and single ventricle circulations, which will be explored further below.

### 3.1. Congenital Heart Disease Associated with Systemic Right Ventricle

In d-TGA, there is atrio-ventriculo concordance but ventriculo-arterial discordance, with the most common van Praagh segmental anatomical classification being {S, D, D}. Systemic venous return flows through the right atrium, right ventricle, and then into the aorta without traversing the pulmonary circulation. Pulmonary venous return flows through the left atrium, left ventricle, and into the pulmonary arteries. The physiology of parallel circulations is incompatible with life without native shunting or surgical remedy [21]. Earlier surgical techniques used atrial baffles to redirect venous inflow to the opposite ventricle, allowing for a deoxygenated systemic venous return to traverse the pulmonary circuit and eject into the systemic circulation thereafter [22]. While this facilitates physiologic oxygenation, a morphologic right ventricle is left to generate systemic pressures far beyond its intended design. Though not all mortality experienced after this atrial switch procedure is due to systemic right ventricular dysfunction, a recent retrospective study showed only 60% of patients who had undergone this procedure were alive after 30 years of follow-up [23].

In congenitally corrected or levo-transposition of the great arteries (l-TGA), there is both atrio-ventricular and ventriculo-arterial discordance, with the most common van Praagh segmental anatomical classification being {S, L, L}. The result is systemic venous return entering the morphologic left ventricle and the pulmonary artery with pulmonary venous return entering the morphologic right ventricle and the aorta. Patients without comorbid cardiac malformations (a minority) can tolerate this physiology into adulthood before detection. Nonetheless, the right ventricle is similarly vulnerable to dysfunction as above given the inherent mismatch in oxygen supply and demand and predominately circumferential instead of longitudinal shortening [24,25]. Patients with l-TGA most commonly have L-ventricular looping, which adds anatomic complexity, as the systemic right ventricle sits on the left side, and the subpulmonic morphologic left ventricle sits on the right side.

Anatomically, the right ventricle is less concave, has a greater degree of trabeculation, possesses a more variable, diffuse subvalvular apparatus and has a thinner ventricular wall than the morphologic left ventricle. Each of these factors presents a challenge to the placement of durable VADs. These anatomic challenges are ameliorated as the right ventricle dilates to assume a more globular morphology in the setting of chronic exposure to systemic afterload.

### 3.2. Durable Mechanical Circulatory Support Considerations in Systemic Right Ventricles

The optimal location for the inflow portion of an MCS device is variable in systemic right ventricles, depending on several factors, including the angle of the patients’ atrioventricular valve inflow, their ventricular arrangement, and mediastinal anatomy. Avoiding the tricuspid subvalvular apparatus and the ventricular septum is paramount. Resection of the trabeculations may be necessary to prevent inflow cannula obstruction [26]. Transesophageal echocardiography is often used to assist with identifying an appropriate inflow cannula location, followed by intraoperative epicardial echocardiography to verify that the inflow cannula position will be free of trabeculations, chordae, and papillary muscles.

Significant tricuspid regurgitation may complicate the MCS placement in the systemic right ventricle, and there are differing opinions among centers as to the need for tricuspid valve repair or replacement at the time of VAD implantation. Where some centers routinely perform bioprosthetic tricuspid valve replacement at the time of VAD implantation, others disregard the tricuspid regurgitation to minimize operative duration [27].

The systemic right ventricle’s position in the anterior chest in d-TGA post atrial switch can create geometric challenges for VAD implantation, with the sewing ring for the inflow cannula typically needing to be on the diaphragmatic surface. Simultaneously, the surgeon must attempt to find a location such that there is an adequate distance from the interventricular septum. Consideration must also be taken such that sternotomy closure does not cause compression of intrathoracic or epigastric structures [28]. A summary of the key considerations is highlighted in Table 1 below.

Despite significant anatomic and physiologic variability in patients presenting with systemic right ventricular failure, case series and registry data demonstrate the feasibility and benefit of durable mechanical support with HeartMate II (Abbott, Chicago, IL, USA) and Heartware devices (Medtronic, Minneapolis, MN, USA) [16,27,29]. However, the only currently FDA approved durable VAD is the HeartMate 3 (Abbott, Chicago, IL, USA). The advantage of the HeartMate 3 over prior generations of devices in the particular circumstance of supporting systemic right ventricles is that the inflow cannula is lower profile and thus less likely to obstruct a heavily trabeculated right ventricle. Additionally, the outflow cannula is larger and theoretically less likely to obstruct when accommodating unique positioning. INTERMACS data show that VADs placed to support a systemic right ventricle have comparable survival to patients with other ACHD phenotypes with a 60% two-year survival, which is comparable to non-ACHD patients supported with durable LVADs in the same time period [16]. Figure 1 below shows an example of HeartMate 3 in a congenitally corrected transposition anatomy.

### 3.3. Single Ventricle Circulations

Patients with a single ventricle palliation with Fontan circulations represent one of the fastest growing populations within ACHD, many of whom experience signs or symptoms of Fontan failure as early as their second or third decades of life [30]. The single ventricle circulation is surgical palliation used in those cases of complex CHD in which dysgenesis of the cardiac mass leaves patients with a single effective ventricle of right, left, or, in many cases, indeterminate morphology. In these cases, complex intercameral communications, valvular disease and/or cardiomyopathy preclude surgical separation of the traditional pulmonary and systemic circulations.

In a series of surgeries, systemic venous return is redirected to the pulmonary arterial circulation without an interposed subpulmonic ventricle. Thus, systemic venous return flows passively into the pulmonary arterial circulation. Pulmonary venous return flows back into the heart, across the atrio-ventricular valve(s), and into the single ventricle to supply blood to the systemic circulation. In this physiology, the transpulmonary flow is driven solely by the gradient between the systemic venous pressure and pulmonary venous pressure [31].

This circulatory circuit is the quintessential preload-dependent physiology, so much so, that iatrogenic shunts are created in the last stage of the palliation to ensure adequate ventricular filling [32,33,34]. Diminished preload may be a consequence of venous insufficiency, obstruction of the cavo-pulmonary anastomosis, increases in pulmonary vascular resistance, or diminution of the gradient between the systemic venous and pulmonary venous pressure with hypovolemia or alternate causes.

### 3.4. Durable Mechanical Circulatory Support Considerations in Single Ventricle Circulations

The decision to pursue durable MCS implantation for patients with single ventricle physiology should only be made after a thorough diagnostic evaluation. In addition to delineating medical and surgical risk factors, the goal of such an evaluation should be to elucidate the predominant mechanism underlying Fontan circulatory failure. Issues, such as symptomatic arrhythmia, baffle obstruction, or significant veno-venous collateral shunting may cause severe symptoms that will not be alleviated by placing a traditional VAD. Indeed, implanting a VAD into a single ventricle circulation with cavo-pulmonary obstruction may lead to preload insufficiency and inflow cannula obstruction due to free wall suck-down.

Patient selection is paramount when considering MCS implantation in the single ventricle circulation. Fontan failure with reduced ejection fraction and Fontan failure with increased end-diastolic pressure are the most likely to benefit, as the VAD can compensate for the circulatory dysfunction. Fontan failure with normal hemodynamics and Fontan failure with abnormal lymphatics are unlikely to benefit from the typical systemic VAD therapies [35,36].

Factors, such as repeated sternotomies, device fit, need for concurrent valve repair, baffle revision, and aortopulmonary and veno-venous collaterals should be carefully considered (See Table 1 above). Three-dimensional imaging technologies have made major strides forward and have received interest as an important tool in surgical planning [37,38]. Virtual reality platforms and 3D printing have been used to simulate MCS implantation in replicas of patient-specific anatomy and can help surgeons better understand the optimal angle for implanting the inflow and outflow portions of MCS devices in patients with widely variable anatomy [36,39].

A unique consideration in Fontan patients as compared to non-ACHD patients is the presence of aortopulmonary collaterals, which can divert 15–30% of the aortic blood flow in Fontan patients and represent a volume load on the ventricle [40]. Two recent case reports of HeartMate 3 implantation had differing conclusions, as one report showed previously small aortopulmonary collaterals that underwent rapid progression with hemodynamic compromise and required coil embolization post-VAD implant while another report showed near resolution of collateral burden after three years of VAD support [41,42]. Both cases, however, highlight the need for higher ventricular assist flows to achieve ventricular unloading.

Patients with single ventricle circulations are also prone to protein losing enteropathy and plastic bronchitis—forms of lymphatic dysfunction felt to be a consequence of chronically elevated systemic venous pressure. Clinical data on tolerance of durable MCS with lymphatic dysfunction is sparse, though there is reason to believe it may increase the risk of adverse events after implantation, given the known associations with systemic inflammation, cytopenias, thrombosis risk, frailty, infection, and tissue edema [43,44].

While limited, the published experience for durable MCS implantation in patients with single ventricle physiology is quite positive, demonstrating successful use as a bridge to transplant and recovery. In 2020, the Advanced Cardiac Therapies Improving Outcomes Network published initial data on its multi-institutional Fontan VAD Physiology Project, examining the outcomes of all patients (pediatric and adult) with single ventricle circulations implanted with a VAD [45]. Thirty-nine patients underwent MCS implants between 2012 and 2019. The HVAD was the most frequently implanted device (54%), though a number of intra- and extra-corporeal devices were used. Most patients in the cohort were either alive on support (10%) or had been transplanted (67%) at the time of the publication. Notably, VAD support for multiple patients spanned greater than one year [46]. There is, thus, a potential for VAD as destination therapy in these patients. Figure 2 below shows an example of HeartMate 3 in hypoplastic left heart syndrome post-Fontan.

While experience with implanting VADs in single ventricle circulations is growing, these devices do not address other forms of Fontan circulatory failure. Thus, there has been growing interest in developing cavo-pulmonary pumps to act as a surrogate for a subpulmonic ventricle. At the time of this review, there are no devices routinely used for this purpose. Intravascular axial and rotary devices in the cavo-pulmonary circuit are under development, as are bio-engineered reservoir pumps [47,48,49] (see Figure 3 below). Berlin Heart has designed an implantable sub-pulmonary support system for patients with Fontan failure as a means to improve end-organ function while awaiting transplant [50]. These devices hold promise for averting untoward sequelae inherent to the Fontan physiology in failing systems, and improving patients’ natural history of the disease.

## 4. Temporary Mechanical Circulatory Support

Experience with temporary MCS with ECMO in ACHD has been published [51], though there are no large studies evaluating protocols to transition patients from temporary to durable MCS. There is growing experience with Impella devices (Abiomed Inc., Danvers, MA, USA). These devices are among the more commonly used in non-ACHD patients in the setting of cardiogenic shock as a bridge to durable MCS/transplant or for periprocedural support during complex procedures. Case reports document their use in various congenital heart lesions, including the biventricular circulation with a systemic left ventricle, the systemic right ventricle and the Fontan circulation (see Figure 4 and Figure 5 below) [52,53,54,55,56,57].

Clinicians must be mindful of the issues that ACHD patients may have with large bore peripheral access given from congenital vascular anomalies and scar tissue from prior surgeries and catheterizations. Furthermore, attention must be paid to specific anatomic complexities such as the reconstructed neo-aorta in patients who have undergone a Damus–Kaye–Stansel anastomosis of the aorta and pulmonary artery or the tricuspid subvalvar apparatus of the systemic right ventricle. Most recently, there has been a report of an Impella RP device used to support the sub-pulmonary circulation as a bridge to transplant in a patient with Ebstein anomaly [58]. The use of advanced imaging and/or 3D modeling with virtual reality simulation has similarly been employed to facilitate these novel interventions.

Given the recent update to the UNOS heart transplant allocation criteria, the use of temporary MCS may help shorten the waitlist time for transplants in ACHD patients. In non-ACHD patients, Impella use has been found to decrease end-organ dysfunction and waitlist times [59,60]. Anecdotally, Impella has been used for multiple months as a bridge to transplant, especially in cases where an operation for a durable VAD implant is thought to be high risk.

## 5. Conclusions

The need for advanced HF therapies in ACHD patients stands to grow substantially within the next decade. Though limitations exist, there is increasing evidence to suggest that durable MCS can be implanted safely and with favorable outcomes in this population. Expansion of durable MCS use in ACHD patients may improve their clinical outcomes as centers gain experience, though each case requires thorough diagnostic evaluation and multidisciplinary discussion between HF, ACHD, and surgical specialists.

## Figures and Tables

**Figure 1 jcm-11-03200-f001:**
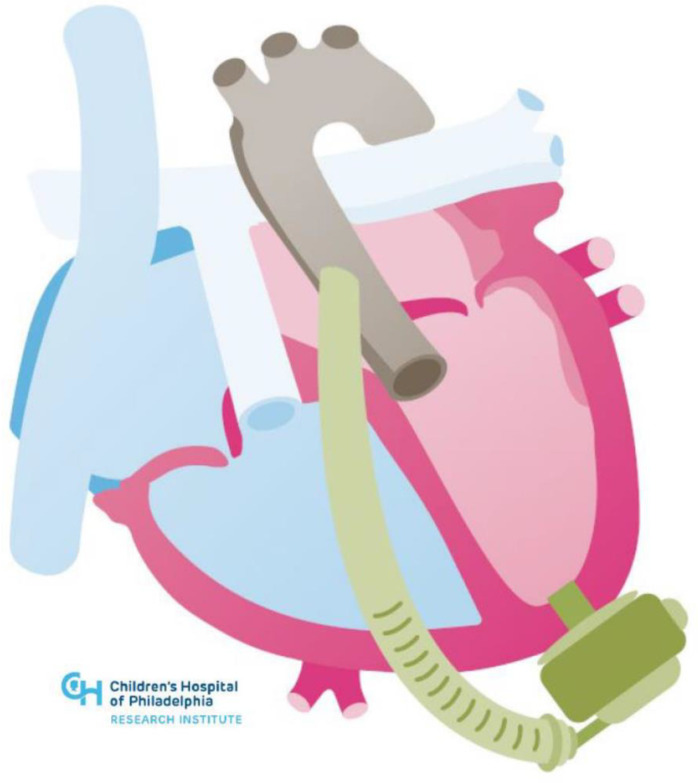
Illustration of HeartMate 3 placement in congenitally corrected transposition of the great arteries.

**Figure 2 jcm-11-03200-f002:**
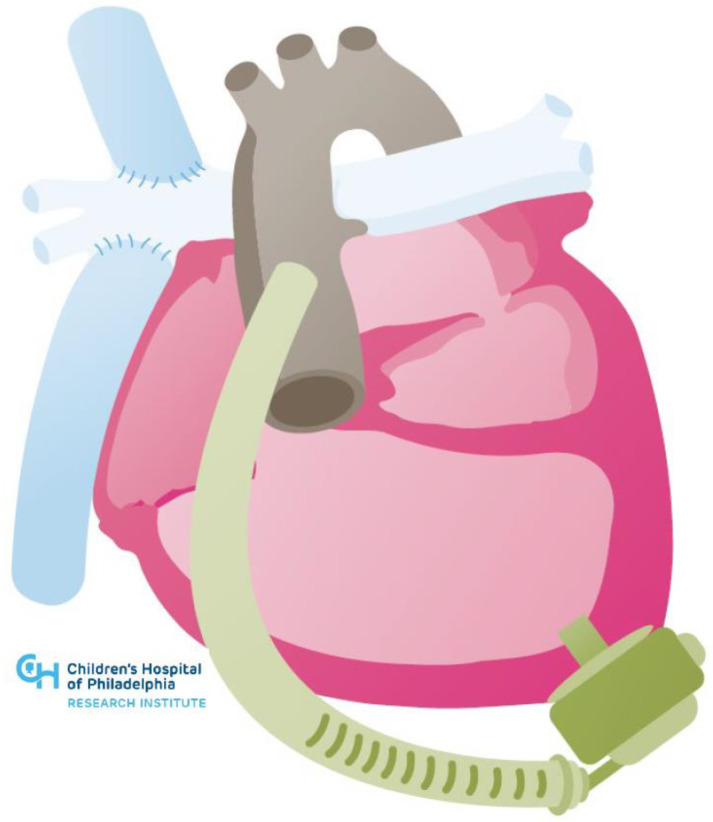
Illustration of HeartMate 3 placement in hypoplastic left heart syndrome post-Fontan.

**Figure 3 jcm-11-03200-f003:**
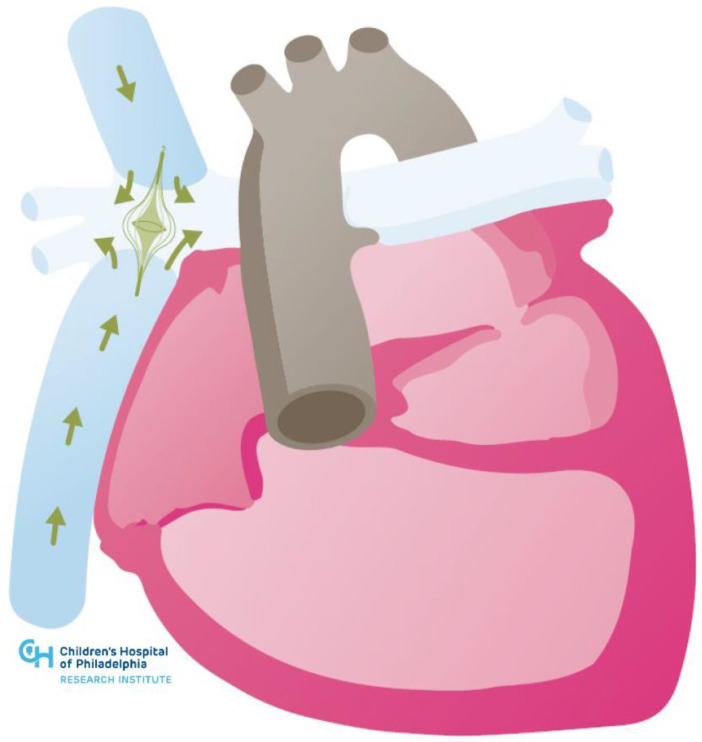
Illustration of cavo-pulmonary assist device in hypoplastic left heart syndrome post-Fontan.

**Figure 4 jcm-11-03200-f004:**
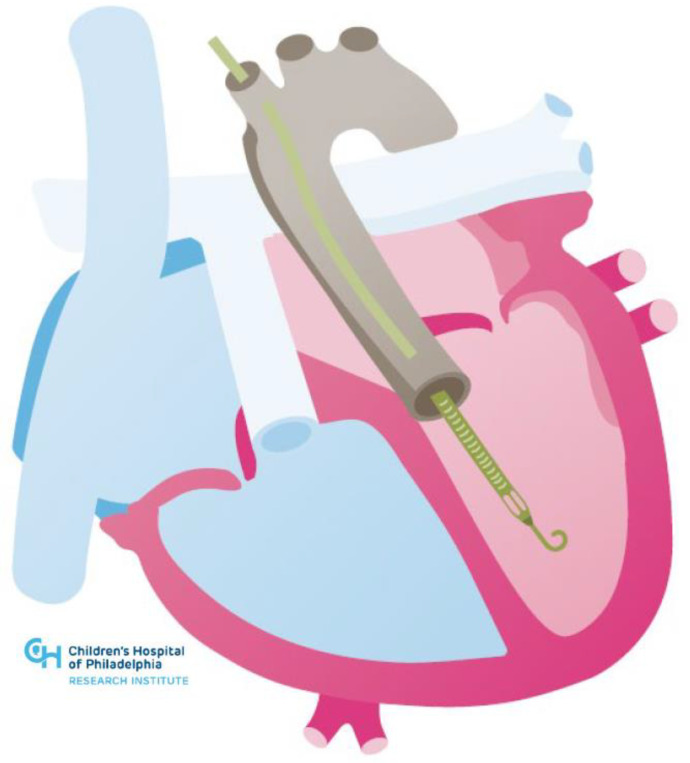
Illustration of Impella in congenitally corrected transposition of the great arteries.

**Figure 5 jcm-11-03200-f005:**
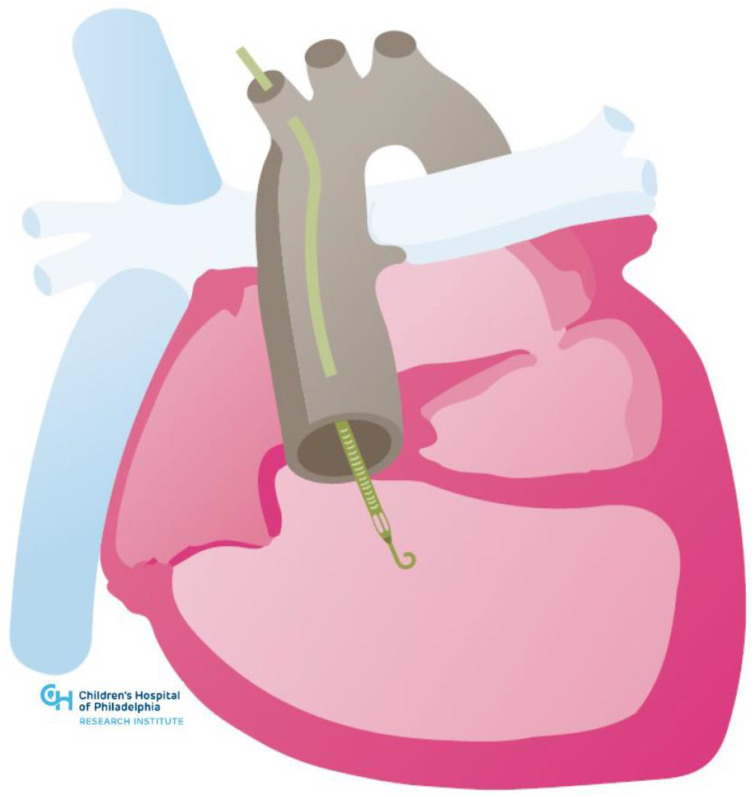
Illustration of Impella in hypoplastic left heart syndrome post-Fontan.

**Table 1 jcm-11-03200-t001:** Summary of Key Considerations for Durable Mechanical Circulatory Support in the Systemic Right and Single Ventricle.

	Systemic Right Ventricle	Single Ventricle
**Cardiac Conditions**	Dextro-Transposition of the Great Arteries or Congenitally corrected Transposition of the Great Arteries	Complex Congenital Heart Disease not amenable to biventricular repair (*e.g., Hypoplastic Left Heart Syndrome, Tricuspid Atresia, Double Inlet Left Ventricle, Unbalanced Atrioventricular Septal Defect*)
**Technical Challenges**	Inflow Cannula Placement (*e.g., trabeculations, papillary muscles*) Concomitant Need for Atrioventricular Valve Surgery or Baffle Revision Abnormal Ventricular Looping Creating Geometric Challenges Scarring and Fibrosis from Prior Sternotomies Achieving Adequate Separation from Ventricular Septum Avoiding Compression of Intrathoracic or Epigastric Structures with Sternotomy Closure	Inflow Cannula Placement (*e.g., trabeculations, papillary muscles*) Concomitant Need for Atrioventricular Valve Surgery or Fenestration Bleeding Risk (*e.g., aortopulmonary or veno-venous collaterals*) Scarring and Fibrosis from Multiple Prior Sternotomies Neo-Aortic Pathology Device Fit
**Management Challenges**	Potential for Inflow Obstruction due to Right Ventricular Anatomic and Geometric Features	Preload Dependency due to Cavopulmonary Flow Comorbidities (*e.g., Fontan-associated liver disease*)
**Device Potential**	Ventricular Assist Devices	Ventricular Assist Devices and Cavopulmonary Assist Devices

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
