# Peer review of "Durable Mechanical Circulatory Support in Adult Congenital Heart Disease: Reviewing Clinical Considerations and Experience"

_jcm, 2022, doi:10.3390/jcm11113200_

Round 1

Reviewer 1 Report

It was a great pleasure to review this paper about „Durable Mechanical Circulatory Support in Adult Congenital Heart Disease: Reviewing Clinical Considerations and Experience“, which is substantially well written and provides all important informations for everybody working and interested in this field. It summarizes the experience of multiple studies and the authors own experience in a very compressed version. The article just lacks tables and figures, but is despite of this absolutely perfect. I personally would like to see at least one table with comprehensive and summarized important informations, which helps to better understand some parts of the subject. One thing I am really desperately missing is some sort of figures like photographs, 3D-CTscans or at least some sort of medical images about supported patients with congenital heart disease. i would be very glad if the authors could provide some sort of the requested additional supplements, because I think they would be very nice to have.

All important issues have been touched:

  • difficult classification and treatment strategies due to lack of studies
  • only 0.1% of patients in the intermacs-registry, thus VAD-implantations are rather very rare or could be regarded as "orphan disease management"
  • unfavourable anatomic and chest features
  • always indication for HTX, rare for destination
  • additional diagnostic informations and further aquired diseases need to be considered (liver function, right heart cath., PVR)
  • univentricular hearts (difficult Fontan support, esp. in patients with chronic PLE) 

Apart of that the above mentioned minor criticism I generally think the paper need some really minor revisions or rather supplements as mentioned above.

Author Response

Response to Reviewer 1 Comments:

It was a great pleasure to review this paper about „Durable Mechanical Circulatory Support in Adult Congenital Heart Disease: Reviewing Clinical Considerations and Experience“, which is substantially well written and provides all important informations for everybody working and interested in this field. It summarizes the experience of multiple studies and the authors own experience in a very compressed version. The article just lacks tables and figures, but is despite of this absolutely perfect. I personally would like to see at least one table with comprehensive and summarized important informations, which helps to better understand some parts of the subject. One thing I am really desperately missing is some sort of figures like photographs, 3D-CTscans or at least some sort of medical images about supported patients with congenital heart disease. i would be very glad if the authors could provide some sort of the requested additional supplements, because I think they would be very nice to have.

All important issues have been touched:

  • difficult classification and treatment strategies due to lack of studies
  • only 0.1% of patients in the intermacs-registry, thus VAD-implantations are rather very rare or could be regarded as "orphan disease management"
  • unfavourable anatomic and chest features
  • always indication for HTX, rare for destination
  • additional diagnostic informations and further aquired diseases need to be considered (liver function, right heart cath., PVR)
  • univentricular hearts (difficult Fontan support, esp. in patients with chronic PLE) 

Apart of that the above mentioned minor criticism I generally think the paper need some really minor revisions or rather supplements as mentioned above.

Response: Thank you for your review and comments. We have added a table and figures to the manuscript.

Reviewer 2 Report

This is a comprehensive review of the literature on the use of mechanical circulatory support (MCS) in adults with congenital heart disease (ACHD). The Authors report both the technical challenges and the data on outcomes in this specific population. Overall, the paper is well written and adequate reference to previous literature is provided. I only have a few suggestions to improve the paper:

  • Albeit not being the main focus of the paper I suggest adding a brief paragraph on heart transplantation in ACHD patients, presenting both technical challenges and prognosis compared to non-ACHD patients.
  • The paper should include at least a table or a figure. I suggest adding a figure presenting the anatomic characteristics of the two entities presented in the paper (namely systemic right ventricle and single ventricle circulations), and also summarizing the main challenges imposed on MCS in these peculiar settings. Furthermore, I suggest adding a Table reporting all the case series with data on the use of MCS in ACHD describing the type of congenital heart diseases treated, the MCS devices employed and, when available, the outcome of these patients.
  • The “2020 ESC Guidelines for the management of adult congenital heart disease” should also be cited.

Author Response

Response to Reviewer 2 Comments:

This is a comprehensive review of the literature on the use of mechanical circulatory support (MCS) in adults with congenital heart disease (ACHD). The Authors report both the technical challenges and the data on outcomes in this specific population. Overall, the paper is well written and adequate reference to previous literature is provided.

Response: Thank you for your review and comments.

I only have a few suggestions to improve the paper:

  • Albeit not being the main focus of the paper I suggest adding a brief paragraph on heart transplantation in ACHD patients, presenting both technical challenges and prognosis compared to non-ACHD patients.

Response: We did not include a brief paragraph on transplantation, as although it is very relevant and related to this topic, this focus issue of the journal specifically was on mechanical support.

  • The paper should include at least a table or a figure. I suggest adding a figure presenting the anatomic characteristics of the two entities presented in the paper (namely systemic right ventricle and single ventricle circulations), and also summarizing the main challenges imposed on MCS in these peculiar settings. Furthermore, I suggest adding a Table reporting all the case series with data on the use of MCS in ACHD describing the type of congenital heart diseases treated, the MCS devices employed and, when available, the outcome of these patients.

Response: We have added a table and figures to the manuscript.

  • The “2020 ESC Guidelines for the management of adult congenital heart disease” should also be cited.

Response: The ESC guidelines have now been cited.